# Deep learning the structural determinants of protein biochemical properties by comparing structural ensembles with DiffNets

Michael D. Ward[1,2], Maxwell I. Zimmerman[1,2], Artur Meller[1,2], Moses Chung [1,2], S. J. Swamidass [3] & Gregory R. Bowman[1,2 ✉]

Understanding the structural determinants of a protein's biochemical properties, such as activity and stability, is a major challenge in biology and medicine. Comparing computer simulations of protein variants with different biochemical properties is an increasingly powerful means to drive progress. However, success often hinges on dimensionality reduction algorithms for simplifying the complex ensemble of structures each variant adopts. Unfortunately, common algorithms rely on potentially misleading assumptions about what structural features are important, such as emphasizing larger geometric changes over smaller ones. Here we present DiffNets, self-supervised autoencoders that avoid such assumptions, and automatically identify the relevant features, by requiring that the low-dimensional representations they learn are sufficient to predict the biochemical differences between protein variants. For example, DiffNets automatically identify subtle structural signatures that predict the relative stabilities of β-lactamase variants and duty ratios of myosin isoforms. DiffNets should also be applicable to understanding other perturbations, such as ligand binding.

[1] Department of Biochemistry & Molecular Biophysics, Washington University School of Medicine, St. Louis, MO, USA. [2] Center for the Science and Engineering of Living Systems, Washington University in St. Louis, St. Louis, MO, USA. [3] Department of Pathology & Immunology, Washington University School of Medicine, St. Louis, MO, USA. ✉email: g.bowman@wustl.edu

A mechanistic understanding of how a protein's sequence determines its structural preferences and, ultimately, its biochemical properties is crucial for advancing our understanding of fundamental biology and for applications in precision medicine and protein engineering. Sequence variations can modulate a protein's biochemical properties in a deleterious manner leading to morbidity and mortality[1,2], or in a manner that can improve a species fitness, e.g., conferring the ability to metabolize new substrates[3]. Moreover, entire protein families with a wide range of functions and biochemical properties emerge after long timescale evolution of protein sequences. In either case, identifying the structural and dynamical differences between protein variants is a powerful means to understand the mechanism that connects a protein's sequence and biochemical properties[4–9]. Streamlining this process would make it easier to infer the behavior of new protein variants, which would accelerate protein engineering and the interpretation of newly discovered variants. Understanding the structural basis for protein function and dysfunction can also accelerate the development of drugs and other therapeutics.

Identifying the structural features that determine the biochemical differences between protein variants is often a difficult challenge, requiring one to consider the entire ensemble of structures that a protein adopts. Techniques like crystallography and cryoEM sometimes reveal dramatic structural differences between protein variants that readily explain their biochemical differences. However, there are also many cases where structural snapshots do not provide a clear explanation for the differences between variants[10], suggesting that one must consider the entire ensemble of thermally accessible configurations these proteins adopt to understand the biochemical differences between them[11–13]. Molecular dynamics simulations can provide access to these ensembles[14]. However, there are many factors that make comparing these ensembles difficult. First of all, proteins have thousands of degrees of freedom that enable them to adopt an enormous number of different configurations[15,16]. Moreover, two ensembles may be highly overlapping, requiring one to identify differences in the probabilities of structural features that are present in both ensembles, rather than simply identifying features that are only present in one ensemble. For example, mutations in the enzyme TEM β-lactamase were found to determine its specificity by modulating the relative probabilities of different structures[12], but all the variants considered had a reasonable probability of adopting any of these structures.

Dimensionality reduction algorithms play a crucial role in dealing with the enormity of conformational ensembles. Many powerful algorithms have been developed and employed successfully, but the utility of each is limited by assumptions that are not universally appropriate. For example, principal component analysis (PCA)[17,18] finds linear combinations of features that retain as much of the geometric variance in the original data as possible, effectively assuming that large structural changes are more important than subtle ones. Unfortunately, there are many cases where this assumption is invalid, as in enzymes where arbitrary motions of a large floppy loop may dwarf subtle but functionally-relevant sidechain motions in the active site. Autoencoders[19] are a more powerful alternative since they consider nonlinear combinations of features. These neural networks learn a low-dimensional projection of data—called the latent space—that is optimized to produce a high-fidelity geometric reconstruction of a protein configuration (Fig. 1). However, like PCA, autoencoders still focus on capturing large geometric variations. Time-lagged independent component analysis (tICA)[20,21] is another common approach. It is similar to PCA but focuses on slowly varying degrees of freedom rather than emphasizing large geometric changes. However, there are many situations where the

conformational changes of interest are fast relative to others (e.g., allostery within the native ensemble that is faster than folding and unfolding of the protein). Another recent approach, VAMPnets[22], combines ideas from autoencoders and tICA to achieve a dimensionality reduction that maps protein structures to metastable states. This allows VAMPnets to capture nonlinearities that tICA cannot, but the assumption that slowly varying degrees of freedom are more important than faster ones is still limiting in many cases. Recent work suggests supervised machine learning algorithms aid in identifying features that distinguish structural states[23]. Here, we explore the idea of integrating supervised machine learning and dimensionality reduction algorithms.

We hypothesized that requiring a dimensionality reduction algorithm to predict the biochemical differences between protein variants would be a powerful means to ensure that it identifies the relevant structural differences without being misled by a priori assumptions. Instead of assuming what type of variation is important (e.g., that large structural changes are more important than smaller ones), such an algorithm would simply assume there are differences between two or more classes of data and then search for features that separate these classes.

To test this hypothesis, we introduce DiffNets, a dimensionality reduction algorithm that uses a self-supervised autoencoder to learn features of a protein's structural ensemble that are predictive of the biochemical differences between protein variants (Fig. 1). While we focus on protein variants, the algorithm should be equally applicable to other perturbations, such as understanding the impact of post-translational modifications and interactions with binding partners. DiffNets takes two inputs: 1) a set of molecular dynamics simulations for each protein variant and 2) the biochemical property of interest (e.g., stability or activity) for each variant. The algorithm then learns a low-dimensional projection (latent space) of the protein structures that is explicitly organized to separate structural configurations based on how closely they are associated with the biochemical property of interest. DiffNets achieve this by combining supervised autoencoders[24] with self-supervision. Supervised autoencoders are multi-task networks. Like standard (unsupervised) autoencoders, they must learn a low-dimensional projection of the data that retains sufficient geometric information to reconstruct the original high-dimensional input (Fig. 1, left). However, supervised autoencoders add the additional requirement that the low-dimensional projection of the data be sufficient to predict a label, in this case one related to the biochemical property of interest. This second requirement forces the dimensionality reduction to dedicate representational power to identifying degrees of freedom that are important for the label instead of focusing exclusively on large structural changes. The classification task can be based on the entire latent space to minimize assumptions, or on a subset of the inputs (e.g., the region around a mutation, as in Fig. 1) to focus attention on critical areas. Self-supervision provides an automated way to deal with the fact that we know the biochemical properties of variants (i.e., their entire structural ensemble), but the association between any specific structure and that biochemical property is unknown. This problem is non-trivial because there is likely to be overlap between ensembles (i.e., structures that are visited by all variants). Therefore, classifying all structures from variants without the property of interest as different than all structures from variants with the property is likely a misleading oversimplification. To overcome this limitation, we present an expectation maximization scheme that iteratively updates training labels to identify a subset of structures that are more probable for variants with the biochemical property of interest while allowing for overlap between the conformational ensembles of different variants.

**Standard Autoencoder**

**Custom DiffNet**

**Fig. 1 Standard autoencoder architecture (left) and an example DiffNet architecture (right).** Autoencoders have an encoder that compress the input data to a bottleneck, or latent, layer and a decoder that expands the latent representation to reconstruct the original input. The DiffNet adds a classification task to the latent space. In the example shown, the input is split into two encoders. One is a supervised encoder that operates on atoms near the mutation (cyan) and must predict the biochemical property associated with a structure. The second encoder is unsupervised and operates on the rest of the protein (blue). The latent layers from these two encoders are concatenated and trained to reconstruct the original input.

To test the performance of DiffNets, we apply them to a set of four TEM β-lactamase variants, which differ by single point mutations, and to a set of eight myosin isoforms. First, we demonstrate how the DiffNet classification task alters dimensionality reduction of protein structures compared to standard autoencoders. Then, we use DiffNets to recapitulate known differences in β-lactamase variants' folded ensembles that are predictive of changes in stability between variants. The relevant changes are geometrically subtle (<1 Å distance change) compared to other motions, and thus, it originally took our group several months to identify them. Therefore, attempting to recapitulate this result is a challenging test case for new methods, such as DiffNets. Finally, we use DiffNets to understand the structural determinants of duty ratio (i.e., the amount of time a myosin protein spends attached to actin) among eight myosin isoforms. This is a difficult test case since small loop motions are critical for determining duty ratio, which is difficult to pick out in large (e.g., ~800 residues) myosin motor proteins. Further, the underlying amino acid sequences of isoforms are highly divergent, so success on this task would demonstrate that DiffNets are applicable to variants with more complex perturbations compared to single-point mutations.

## Results

**The DiffNet architecture**. The DiffNet architecture is based on an autoencoder, which is a deep learning framework commonly used for dimensionality reduction[4,25–36] (Fig. 1). Like standard autoencoders, DiffNets connect an encoder and decoder network to compress and reconstruct input data, respectively. In our case, the input is protein XYZ coordinates (C,CA,N,CB) from a simulation frame, which are whitened for normalization (see methods). First, the encoder network transforms the input to progressively reduce the dimensionality of the input to a bottleneck layer, called the latent space. Then, the latent space vector is used as input to the decoder network that attempts to reconstruct the original input. Mechanically, both the encoder and decoder operate via successive matrix multiplications and non-linear activation functions. DiffNets (and autoencoders) are initialized with random matrix multiplications, and the network improves by iteratively tuning the matrix values (weights) by training across many examples. Concretely, the weights are trained to minimize a loss function that measures the difference between the input and output of the model, called the input reconstruction

error. Ultimately, if a DiffNet (or autoencoder) can compress and then reconstruct the original input with high accuracy, this implies that the low-dimensional latent space vector retains the salient features that describe the input.

Inspired by supervised autoencoders[24], DiffNets augment autoencoders with a loss function that measures how accurately the latent space vector performs a user-defined classification task (e.g., did the protein structure come from a wild-type or variant simulation?). Therefore, DiffNets must learn weights that simultaneously minimize protein reconstruction error and classification error. The constraint to minimize protein reconstruction error enforces that the low-dimensional representation of data retains a structural basis, and the classification constraint is designed to reconfigure the latent space such that data points are separated to highlight differences between datasets (e.g., biochemical differences between protein variants). While supervised autoencoders have been previously used as a way to obtain better performance on a classification task[24], we use the classification task to learn a more interpretable low-dimensional projection of data. Additionally, we propose an expectation maximization scheme such that classification labels are updated between DiffNet training epochs. This self-supervision provides an automated way to deal with the fact that we know the biochemical properties of variants (i.e., their entire structural ensemble), but the association between any specific structure and that biochemical property is unknown.

The DiffNet architecture can be split to focus the classification task on a region of interest within a protein. If there is a region of interest known a priori (e.g., region around a mutation, or an enzyme active site) the input may be split into two encoder networks. In this case, only the encoder with inputs from the region of interest performs a classification task, then the latent spaces from each encoder are concatenated for input to the decoder (see Fig. 1). This split architecture guides a DiffNet to search in the region of interest to find differences between variants. This is a reasonable default to use when studying single point mutations as the region of a mutation is root of differences between variants. Moreover, classifying based on a region of interest does not preclude the identification of relevant distal structural differences between variants. If a mutation causes biochemically relevant differences at distal regions then these regions are inherently linked to the state of the region of the mutation and, thus, are implicitly linked to the classification task.

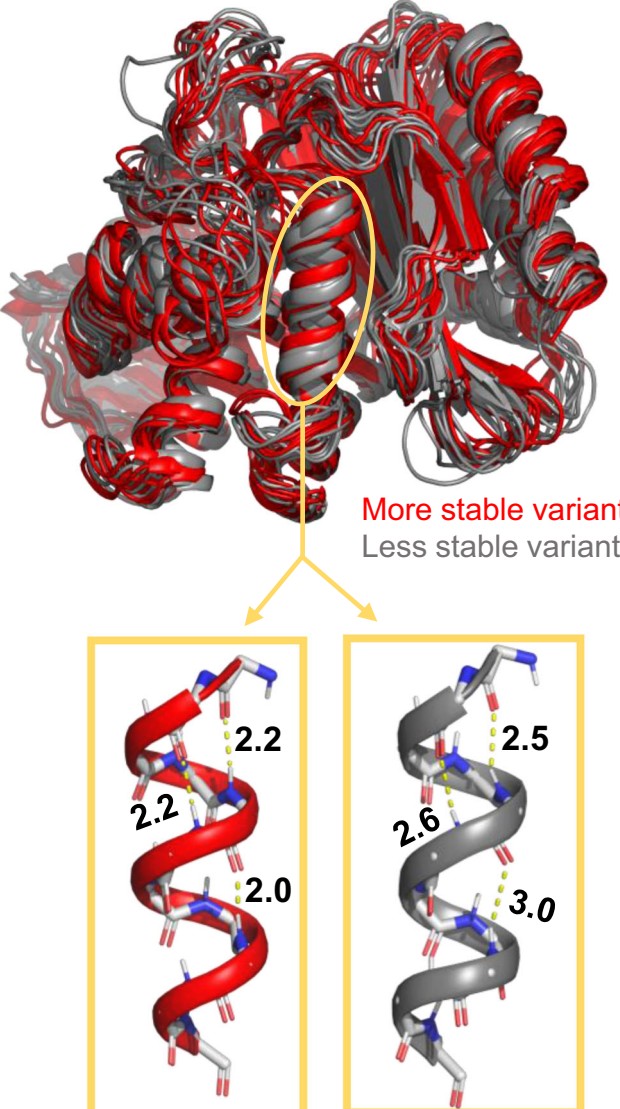

**Fig. 2 Structural configurations sampled from molecular dynamics simulations of wild-type TEM β-lactamase (gray) and an M182T variant (red) that is far more stable.** Helix 9 is circled in yellow and shown below in a compact configuration (left, red) and a more extended configuration (right, gray). Hydrogen bond distances are shown in Ångstroms.

**The classification task reorganizes the latent space to emphasize biochemically-important structural features.** Dimensionality reduction algorithms are only helpful for identifying differences between two classes of data if the two classes of data are separated in the latent space. Unsupervised autoencoders learn a latent representation of data that focuses on large geometric variations, so structures with large geometric differences are separated, while structures with subtle differences are close together. As a result, if biochemical differences between protein variants are related to subtle geometric changes, then the variants will be highly overlapping in the latent space and thus, the autoencoder will fail to provide a useful way to distinguish variants. We hypothesized that augmenting a standard autoencoder with a classification task, as with DiffNets, would reorganize the latent space to highlight relevant differences between datasets, even if they are subtle structurally.

In order to test this hypothesis, we applied DiffNets and autoencoders to a set of variants of the enzyme TEM β-lactamase. β-lactamase is an enzyme that confers bacteria with antibiotic

resistance by metabolizing β-lactam drugs like penicillin[37]. Bacteria are quick to evolve new variants of TEM that have activity against new drugs, but these mutations are often destabilizing, so compensatory mutations evolve to restore stability[38–40]. M182T is one stabilizing mutation that frequently appears in clinical isolates[41,42]. While crystal structures of the wild-type and M182T proteins had been solved, comparing them did not provide a conclusive mechanism for stabilization capable of predicting the impact of other variants. Recently, our group combined simulations, NMR experiments, and x-ray crystallography to demonstrate that compaction of helix 9 is a structural signature that distinguishes more stable variants (like M182T) from less stable ones (Fig. 2). This compaction is associated with stronger h-bonds along helix 9 that stabilize this secondary structure element. Helix 9 is part of a crucial interdomain interface, so stabilizing it ultimately stabilizes the native state relative to an intermediate where one domain is at least partially unfolded. Importantly, this helix compaction includes distance changes of less than 1 Ångstrom between hydrogen bonding partners. Given that this is geometrically subtle compared to nearby loop motions, we expect that compact and extended helix states will not be well separated in the latent space of a standard autoencoder. However, we do expect that a DiffNet trained to classify compact and extended helix states will learn a latent space that separates these states.

To evaluate if the DiffNet classification layer alters the latent space in a way that helps identify differences between two classes of data, we compared the latent space of DiffNets to the latent space of unsupervised autoencoders after training on a dataset that includes two classes of data distinguishable by a subtle difference in helix 9 compaction. From the original set of 650,210 structures (from wild-type and M182T simulations) we curated a dataset of 178,402 simulation frames from wild-type and M182T simulations where half of the frames have a compact helix 9 (helix compaction criteria described in Methods) and half have a more extended helix. Then, we trained DiffNets and unsupervised autoencoders using a split architecture described in the methods and visualized in Fig. 1b. The DiffNets and autoencoders we trained were identical, except the DiffNet has an additional output layer such that it has to classify helix 9 as compact or extended in addition to reconstructing protein structures. The classification labels are not updated with expectation maximization in this case. This dataset was selected specifically to evaluate how the classification task of the DiffNet alters the dimensionality reduction compared to a standard autoencoder. In a normal setting we would not have a priori knowledge about the importance of helix 9 compaction. However, this is an important test to determine if adding a classification task can reorganize the latent space to highlight differences between datasets, which is a property that DiffNets will ultimately need to identify differences between variants.

Requiring DiffNets to perform a classification task in tandem with dimensionality reduction successfully reconfigures the latent space to disentangle compact helix configurations from more extended helix configurations. First, we note that DiffNets and unsupervised autoencoders have similar ability to reconstruct protein structures (~1 Ångstrom error - see Fig. 3) using as few as three latent variables and as many as fifty, which is in line with another study reporting autoencoder reconstruction error[27]. To compare latent spaces, we analyze a split architecture that has twenty-five latent variables including three in encoder A (which receives input including helix 9 and performs the classification task in the DiffNet) and twenty-two in encoder B (takes input from the rest of the protein). This architecture provides a low reconstruction error (<1 Ångstrom) and few enough latent variables so that all dimensions in encoder A's latent space can be

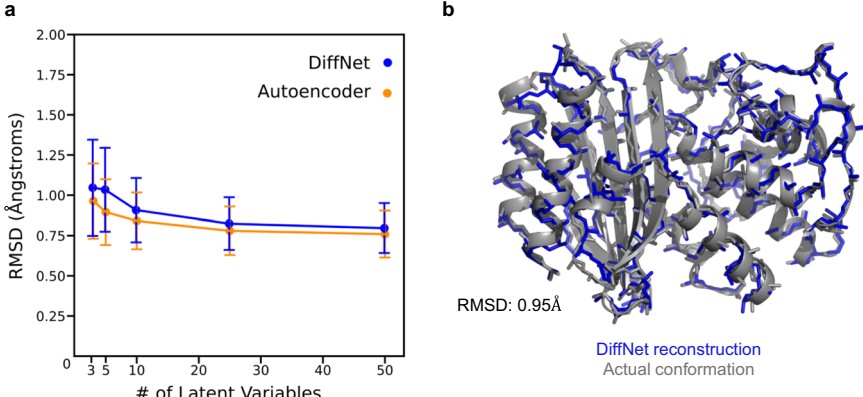

**Fig. 3 Autoencoders and DiffNets can both compress protein structures and then reconstruct them. a** Reconstruction error plots showing the root-mean-square deviation (RMSD) between a protein structure from simulation and the corresponding protein structure generated by unsupervised autoencoders (yellow) or DiffNets (blue). One of every ten structures from wild-type and M182T simulation data was used ($n = 65{,}210$) and the standard deviation is shown with error bars. **b** Structure representing the difference between a structure generated by the DiffNet (blue) vs. the actual conformation from simulation (gray) when training with 3 latent variables.

visualized. In the unsupervised autoencoder, simulation frames of compact and extended helices are overlapping in encoder A's latent space (Fig. 4a). This demonstrates that training an unsupervised autoencoder on two classes of data does not necessarily yield a latent representation that provides any insight into how the two classes of data are different. To explore this point further, we held the autoencoder's latent space constant and then trained it to classify whether a structure has a compact or extended helix 9 (i.e., performed logistic regression). The resulting receiver operating characteristic (ROC) curve, which measures classification performance, shows a classification performance similar to random guessing (area under the curve [AUC] = 0.54) providing further evidence that the latent representation does not help distinguish the two classes of data. In contrast, the DiffNet encoder A latent space clearly separates the two classes of data (Fig. 4a) and has excellent performance classifying compact and extended helix states (AUC = 0.91, Fig. 4b). This result demonstrates that adding a classification component to the learning task provides a powerful means to learn a low-dimensional representation that highlights crucial differences between datasets. It follows that DiffNets trained with a classification task that must predict a biochemical property should learn a low-dimensional representation of data that highlights structural features that are predictive of biochemical differences between protein variants.

**Self-supervised DiffNets learn structural signatures associated with protein stability.** While the classification task can help DiffNets learn a useful dimensionality reduction, realizing this potential is non-trivial because we know the biochemical properties of variants (e.g., their entire structural ensembles) but not individual structures. The simplest approach to providing these classification labels would be to assign ones to structures from simulations of variants with the biochemical property of interest and zeros to structures from simulations of variants without the property. However, it is likely that variants fall on a continuum rather than having a biochemical property or not, that their conformational ensembles overlap, and that only a subset of conformations are relevant for determining the property of interest.

This problem is similar to multiple instance learning. During multiple-instance learning, learners are given bags of training examples where each bag is labeled negative, indicating that the bag contains all negative examples, or positive, indicating that

there are at least some positive examples in the bag. The learner then must figure out how to label all of the individual instances as positive or negative by identifying features that are consistent in positive bags, but absent in negative bags. This is similar to our situation where we know the biochemical property of each protein variant (i.e., negative bag or positive bag), but we do not know if a given structural configuration is associated with a biochemical property, or inconsistent with a biochemical property.

We propose a self-supervised approach for learning the relationship between individual structures and the biochemical property of interest using an iterative expectation maximization algorithm based on work from Zaretski et al[43]. Expectation maximization is a statistical method that allows the parameters of a model to be fit, even when the outputs of the model cannot be observed directly in the training data[44] (i.e., when they are hidden). In our case, the hidden variables are labels for each structure that specify the probability that a structure is associated with the biochemical property of interest. These labels are initially set to ones for all structures from variants with a given biochemical property (e.g., more stable β-lactamase variants) and zeros for variants without that property (e.g., variants with lower stabilities). Then the expectation maximization algorithm iteratively alternates between a maximization step and an expectation step to identify a self-consistent set of labels. During the maximization step, a DiffNet is trained to predict the current labels for each structure. Then, the expectation step refines the training labels by computing the expected values of the labels, $y$, using the output from the DiffNet, $\hat{y}$, conditioned on constraints about what fraction of structures from each variant we expect to be associated with the property of interest. This constraint provides a way to enforce that more high probability values are assigned to structures from variants with the biochemical property. The expectation is the probability-weighted average of all binary realizations of binomial distributions parameterized by $\hat{y}$, excluding binary realizations that do not meet the constraint. Formally, we update training labels as,

$$y_i = E[\hat{y}_i | S_L \le \hat{y}_r \le S_U] \tag{1}$$

$$= P(\hat{y}_i \text{ is } 1) * \left( \frac{P(S_L - 1 \le \hat{y}_r - \hat{y}_i \le S_U - 1)}{P(S_L \le \hat{y}_r \le S_U)} \right) \tag{2}$$

where $y_i$ is the updated label for each individual frame, $\hat{y}_i$ is the DiffNet output, $S_L$ and $S_U$ are the lower and upper bounds on

**a**

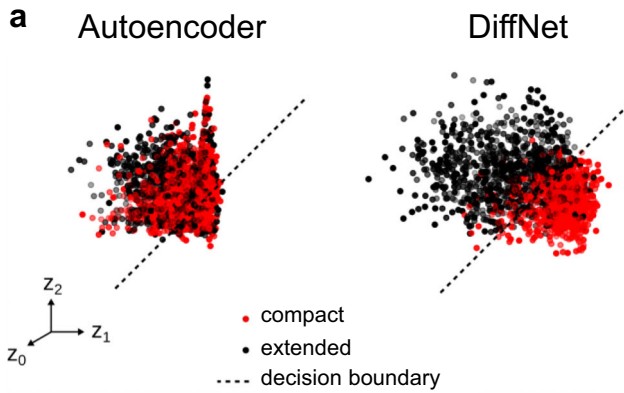

Autoencoder DiffNet

- compact
- extended
- - - - decision boundary

**b**

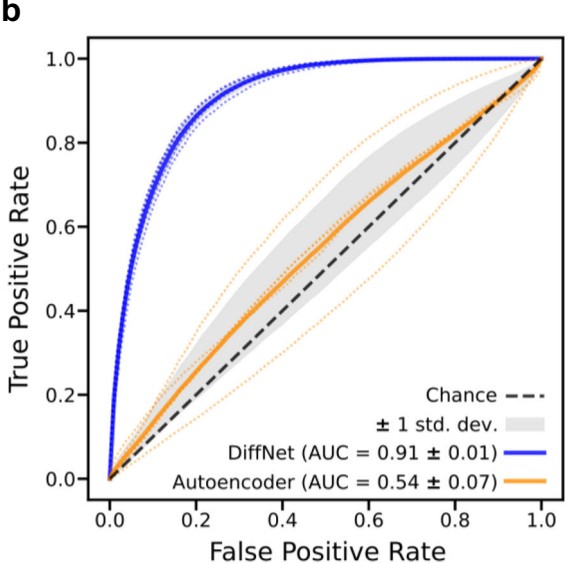

**Fig. 4 Adding a classification component to the learning task (as in DiffNets) results in a latent representation that separates different datasets more clearly than an unsupervised autoencoder. a** Simulation frames that have a compact helix (red) and an extended helix (black) are projected onto the three-dimensional latent space learned by an unsupervised autoencoder (left) and a DiffNet (right). The decision boundary (black dotted line) indicates the plane that each neural network uses to separate compact helix states from extended helix states. **b** Receiver operating characteristic (ROC) curve showing the average classification performance of the unsupervised autoencoder (dark yellow) and the DiffNet (dark blue) as well as the performance for each of the 5 folds of cross validation (faded dotted lines). Mean area under the ROC curve (AUC) is shown in the bottom right corner with the standard deviation across the 5-folds of training.

how many conformations in a batch are associated with the biochemical property, $\hat{y}_r$ is the sum of the binary outcomes of a batch which contains conformation $i$, $P(S_L - 1 \leq \hat{y}_r - \hat{y}_i \leq S_U - 1)$ is the probability that the number of conformations in a batch is within the limits if conformation $i$ is ignored, and $P(S_L \leq \hat{y}_r \leq S_U)$ is the probability that the number of conformations in a batch is within the limits, including conformation $i$. Ultimately, the desired outcome is that the expectation maximization algorithm redistributes training labels from all 0 s and 1 s for simulation frames of variants without and with a biochemical property, respectively, to values that indicate the probability that a given structural configuration is associated with the biochemical property of interest. This mechanism is self-supervised since the

training labels are learned by the algorithm, rather than explicitly curated.

To test this approach, we trained a self-supervised DiffNet to identify structural preferences that distinguish two highly stable β-lactamase variants (M182T and M182S) from two less stable variants (wild-type [WT] and M182V). In this case, the DiffNet receives no a priori information about features, like helix 9 compaction, that are associated with increased stability in M182T and M182S. If self-supervision of DiffNets works as expected, then training should produce a latent space where it is easy to identify the structural features that are associated with the stability of M182T and M182S, relative to WT and M182V. For example, we expect to see structural configurations with a compact helix 9 in one region of the latent space and structures with a more extended helix elsewhere. Beyond helix compaction, DiffNets may even capture additional structural features that were missed in our previous manual analysis. To evaluate if the DiffNet learns these biochemically relevant structural differences between variants, we trained a DiffNet on 6.5 μs of simulation data for each variant: M182T, M182S, WT, and M182V. All frames from M182T and M182S (highly stable variants) were initially assigned classification labels of 1, and simulation frames from M182V and WT were initially assigned 0 s. During the expectation maximization procedure, we calculate the expected values (updated labels) conditioned on the constraint that 0–30% of less stable variants frame are likely to be stabilizing, and 60–90% of frames for highly stable variants. In general, it should be sufficient to base bounds on qualitative a priori knowledge rather than precise, quantitative information. In this case, we chose these bounds as a way to allow overlap between ensembles, but still provide a clear signal to distinguish more and less stable variants. Empirically, we find that DiffNets are robust across a wide range of bounds (Supplementary Fig. 1).

Expectation maximization aids the DiffNet in learning a low-dimensional representation that accurately identifies that helix 9 compaction is associated with highly stable variants. First, we trained two supervised autoencoders (one with and one without expectation maximization) and compared the distribution of output classification labels. Without expectation maximization, almost all structures from more stable variants have output labels close to 1, and structures from less stable variants have output labels close to 0 (Fig. 5a). This is at odds with the fact that there is structural overlap between the ensembles. It indicates that the supervised autoencoder essentially memorizes which ensemble each structure comes from instead of learning a useful association between individual structures and stability. In contrast, when expectation maximization is applied the output labels span the full spectrum from 0-1 for each variant (Fig. 5b), which is consistent across a wide range of expectation maximization bounds (Supplementary Fig. 1). Moreover, as the labels increase from 0 to 1, helix 9 compaction smoothly decreases, which indicates that DiffNets learn a latent space with a continuum of structures that are less/more closely associated with stability (Fig. 5c). Without expectation maximization, the extreme labels (i.e., 0,1) track well with helix stability, but structures labeled between 0.1 and 0.9 do not show a clear trend of helix compaction.

Using DiffNets to predict on a variant outside of training provides further support that expectation maximization aids in learning structural features associated with stability. We compared each model's ability to predict the stability of a less stable variant not seen during training (M182N), and we find that this prediction is improved when expectation maximization is applied (Supplementary Fig. 2). This suggests expectation maximization helps DiffNets hone in on biochemically relevant structural features, and that DiffNets could be used as a predictive tool.

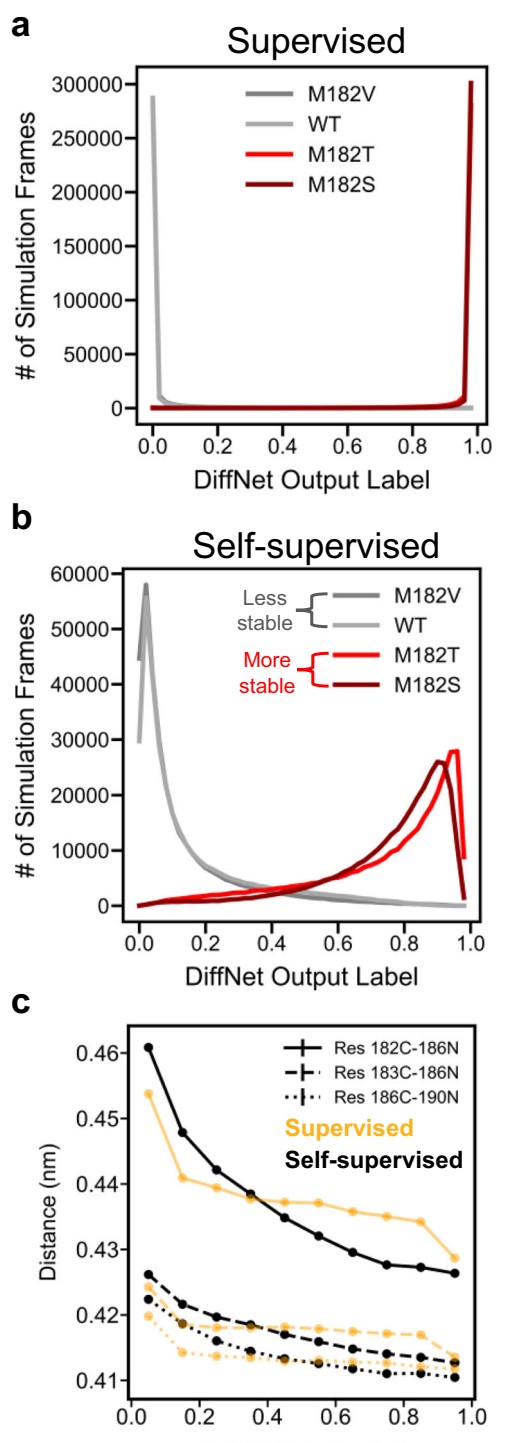

**a** Supervised

**b** Self-supervised

**c**

**Fig. 5 Self-supervision improves the DiffNet's ability to organize structural configurations based on their biochemical property.** Histogram showing DiffNet output labels across all simulation frames from M182T and M182S (red – highly stable variants in training set) versus WT and M182V (gray – less stable variants in training set) for a supervised autoencoder (**a**) and a self-supervised autoencoder (**b**). **c** Three key hydrogen bond lengths in helix 9 as a function of the DiffNet output label ($n = 1,300,420$) (yellow – supervised, black – self-supervised), which ranges from zero for structures associated with low stability to one for structures associated with high stability. The distances are between the carbonyl carbon of the $i$th residue and the nitrogen of the $(i + 4)$th residue. Standard error bars are not visible since the standard error is smaller than scatter points.

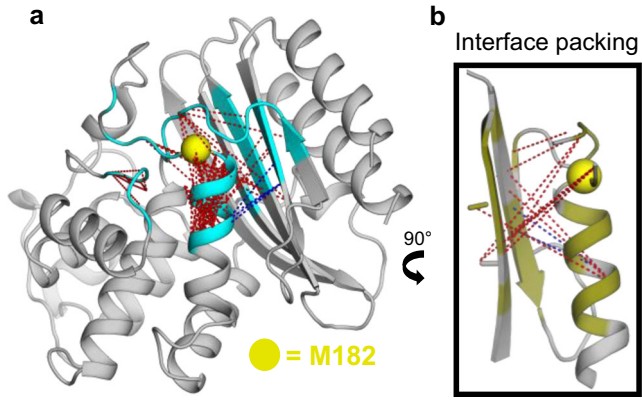

**Fig. 6 Visualization of the features that DiffNets find important for increased stability of M182T and M182S variants. a** Crystal structure of TEM β-lactamase (PDB ID: 1JWP) overlaid with dotted lines that indicate distances between two atoms that change in a way that is strongly correlated with an increased DiffNet output label. Red indicates the atoms move closer together as the output label increases, blue indicates atoms moving away from each other. The mutated residue is highlighted with a yellow sphere. Protein atoms are colored cyan if they are near the mutation, which indicates that they were included in the classification task and considered for the distance correlation calculation. **b** Rotated inset of **a** showing DiffNet predicted packing at the interface of helix 9 and the adjacent β-sheet. Residues with chemical-shift perturbations in M182S relative to wild-type are shown in deep olive.

However, we caution that autoencoders will fail anytime they are applied to data that is highly dissimilar from the training set, so a DiffNet will not perform well on new variants that visit conformations not visited in the training set. Future studies would be necessary to optimize DiffNets for prediction and should be evaluated against related methods such as by Riesselman et al.[45].

While many deep learning approaches are criticized for their lack of interpretability, the DiffNet architecture provides opportunities to understand what the network learned, which provides biophysical insight. To automate DiffNet interpretation, we measured all inter-atom distances within 1 nm of the mutation using 2000 cluster centers calculated from all simulations and then measured the linear correlation between each distance and the DiffNet output label. We plot the top 1% of distances correlated with the DiffNet output label to visualize the conformational changes that the DiffNet views as important for distinguishing stable variants from less stable variants. Encouragingly, the distance correlations strongly point to helix 9 compaction as an important feature of more stable variants (Fig. 6). While the helix compaction is striking, DiffNets also captured other trends that our previous computational analyses did not detect. For example, our NMR data suggested that the packing between helix 9 and adjacent β-sheet differs in more stable vs less stable variants[5], but our computational analysis did not detect a clear trend. On the same simulation dataset, the DiffNet clearly learns that this interface becomes more tightly packed for more stable variants (Fig. 6). Specifically, the DiffNet analysis suggests more stable variants have tighter packing at the helix 9 and β-sheet interface (Fig. 6b). Often times the important features that distinguish protein variants can be complicated and, therefore, easily missed even with months of analysis. DiffNets can learn complicated features and help automate the process of identifying biochemically relevant structural features that distinguish protein variants.

**DiffNets works for other proteins and more divergent sequences.** In order to explore the broad applicability of DiffNets, we also trained a self-supervised DiffNet to identify structural features that distinguish high duty myosin motor domains from low duty myosins. Myosins are a ubiquitous class of motor proteins that perform an extraordinary diversity of functions despite sharing a common mechanochemical cycle[46]. In order to perform roles as diverse as muscle contraction and intracellular transport, myosins have precisely tuned their duty ratios, or the fraction of time a myosin spends attached to actin during one full pass through its mechanochemical cycle. Recent work from Porter et al.[47] suggests that the conformational ensemble of the active site P-loop encodes duty ratio through the balance of nucleotide favorable and unfavorable states. Specifically, low duty motors have an increased propensity to adopt a P-loop "up" state, where the S180 carbonyl group sterically occludes nucleotide binding, whereas high duty motors favor a "down" state, where the P-loop is nucleotide compatible (see Fig. 7b).

We trained a DiffNet using molecular dynamics simulation data from the active sites of four low duty motors and four high duty motors to see if we could recapitulate the trend between P-loop dynamics and duty ratio (Fig. 7a). Importantly, this test case is especially challenging because it includes eight different proteins with a low degree of sequence conservation in the area of interest (i.e., 34% of residues were perfectly conserved within the training area). Low duty motors were given an initial label of zero and high duty motors were initially given a label of one.

A DiffNet trained to distinguish high and low duty myosin motors substantiates previous work that identified P-loop

dynamics to be important for distinguishing these myosins. To determine if a DiffNet captures the importance of P-loop "up" and "down" states, we examined structures with low and high DiffNet output labels (i.e., predicted low and high duty respectively) from a single isoform. We saw a consistent trend in the orientation of the S180 carbonyl group, where structures with high DiffNets labels are in the "down" orientation and structures with low labels are in the "up" orientation (see Fig. 7b). This indicates that the DiffNet correctly learned that high-duty motors are more likely to be in the "down" state and vice-versa. To more precisely quantify this trend, we examined the correlation between DiffNet output labels and nucleotide compatibility (as defined previously[47]) for all frames. We find that as the DiffNet output labels increase (i.e., shift from low duty to high duty), there is a concurrent increase in the ratio of nucleotide favorable:unfavorable states (Fig. 7c).

Automated interpretation of a DiffNet captures the importance of P-loop dynamics and suggests other order parameters that may distinguish high and low duty myosins. Similar to Fig. 6, we calculate the correlation between interatomic distances and DiffNet output labels for all 139,129 distances around the active site (Fig. 7a) and then project the top 100 correlated distances onto the structure (Fig. 7d). This analysis finds 78 distances between the P-loop and the loop connecting the third beta sheet with the SH2 helix (referred to as the β3-SH2 loop), again highlighting that the DiffNet learns that P-loop dynamics are important for discriminating high and low duty motors. We compared this result to a model trained without expectation maximization and find that expectation maximization improves

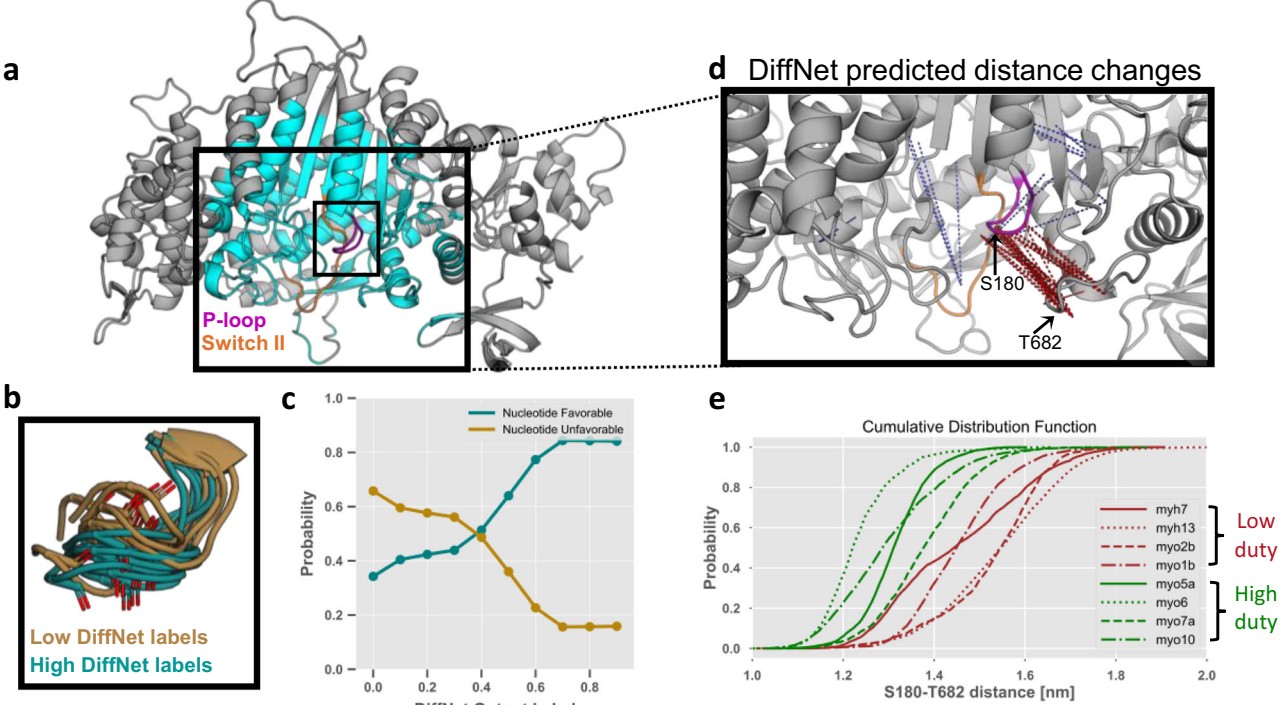

**Fig. 7 DiffNets capture the importance of P-loop motions in distinguishing high and low duty myosin motor proteins. a** Structure of a myosin motor protein (PDB ID: 4PA0) showing the DiffNet classification region (cyan), the P-loop (magenta), and Switch-II (Orange). **b** Twenty states predicted by the DiffNet as high duty (teal) and low duty (dark gold). Predicted high duty states are mostly in a nucleotide compatible, P-loop "down" conformation and vice-versa for predicted low duty states. **c** Percentage of nucleotide favorable (teal) and unfavorable (dark gold) states as a function of the DiffNet output label, measured with 10 equally spaced bins with labels spanning 0-1. Most structures with low Diffnets labels are nucleotide unfavorable, and vice-versa. **d** Inset of the myosin active site. Dotted lines indicate distances between two atoms that change in a way that is strongly correlated with an increased DiffNet output label. Red indicates the atoms move closer together as the output label increases, blue indicates atoms moving away from each other. **e** Cumulative distribution function showing the distance between S180 (P-loop) and T682 (β3-SH2 loop). Probabilities come from a previously published MSM[47]. This distance clearly separates high and low duty motors (green and light brown, respectively) as predicted by the DiffNet in **d**.

the quality of this analysis. Specifically, changes in Ser180 are strongly detected when expectation maximization is applied, but without expectation maximization these changes are not detected at all (Supplementary Fig. 3). The DiffNet also infers that high duty motors are more likely to occupy states where the P-loop is near to the β3-SH2 loop, and indeed this finding is confirmed using previously published Markov State Models of the motor domains (see Fig. 7e). Since the β3-SH2 loop is below the P-loop, this provides further evidence that the DiffNet is correctly learning that high duty motors prefer the "down" state. While this order parameter is the predominant feature of this analysis, the DiffNet suggests that other distances may be important for distinguishing high and low duty motors. In particular, there are two residues on switch-II with distances that are strongly correlated with the DiffNet label indicating that conformational changes in switch-II may be important for determining the duty ratio, which is consistent with previous findings[48] (see Supplementary Fig. 4).

## Discussion

We have introduced DiffNets, a deep learning framework for identifying the structural signatures that are predictive of biochemical differences between protein variants from molecular dynamics simulations. Such simulations contain valuable information about the structural mechanisms that determine proteins' biochemical properties. However, extracting this insight is often difficult because of factors like the high dimensionality of the spaces involved and overlap between the structural ensembles for different variants. Our results suggest that self-supervised DiffNets learn a low-dimensional latent representation of protein structures that separates them based on their association with biochemical properties, such as higher or lower stability. This success relies on two key innovations. First, performing dimensionality reduction simultaneously with a classification task helps yield a latent representation that organizes protein structural configurations based on their association with biochemical properties. Second, challenges with labeling each structure with a biochemical property can be overcome using an expectation maximization scheme inspired by multiple instance learning.

As a proof of principle, we demonstrated that DiffNets automatically identify structural changes that explain biochemical differences between variants in several systems including β-lactamase and myosin proteins. Success identifying helix 9 compaction (<1 Å) as an important distinguishing factor between β-lactamase variants demonstrated that DiffNets finds biochemically relevant structural features even if they are geometrically subtle relative to other structural fluctuations in the protein. Success identifying the importance of P-loop dynamics for determining the duty ratio across myosin isoforms demonstrated that DiffNets is generalizable to large proteins (~800 residues) with low sequence conservation. Looking ahead, we expect the same architecture to be applicable to other perturbations, such as post-translational modifications or the presence/absence of a binding partner.

While these results are promising, future work can be done to expand the utility of DiffNets further. For example, the DiffNet architecture is not translationally, nor rotationally, invariant, which means the results depend on the quality of the initial alignment of simulations. Future work exploring equivariant architectures may improve DiffNets. Additionally, the current study included an abundance of data, so there were no optimizations for working with small datasets. It is yet to be seen how well DiffNets performs on smaller (sub-microsecond) datasets. Lastly, after training a DiffNet it is possible to use the model to predict the biochemical property of a variant for which the

property has not been determined experimentally. Toward this end, we showed that DiffNets accurately classified the stability of a β-lactamase variant (M182N) that was not seen during the training. However, accurate predictions will require that the variants of interest have high conformational overlap, and future studies are required to optimize a model for this task.

## Methods

**Molecular dynamics simulations.** All molecular dynamics simulation data were generated in previous manuscripts by Zimmerman et al.[5] and Porter et al.[47] Briefly, all simulations were run with Gromacs 5.1.1 at a temperature of 300 K using the AMBER03 force field with explicit TIP3P solvent[49,50]. β-lactamase simulations were initialized from the TEM-1 β-lactamase crystallographic structure (PDB ID: 1JWP)[39] and ran at 300 K using the AMBER03 force field with explicit TIP3P solvent[49,50]. Each variant, wild-type, M182V, M182T, M182S, and M182N was simulated for 6.5 μs including 4 μs of FAST-RMSD adaptive sampling[51] and 2.5 μs of conventional sampling. Conformations were stored every 20 ps. Myosin simulations were performed mostly on Folding@Home[52] to obtain ~2 ms of total sampling across four low duty (MYH13, MYH7, MYH10, and MYO1B) and four high duty motors (MYO5A, MYO6, MYO7A, and MYO10), where the initial structures were built from homology models in SWISS-MODEL[53] using the 4PA0[54] as a guide template structure.

**DiffNet model.** DiffNets are neural networks with a supervised autoencoder architecture (as shown in Fig. 1). These models take as input a vector of features that describe a protein structural configuration and output a score which indicates how closely a structure is associated with a certain biochemical property, as well as, a vector that matches the input vector (i.e., reconstructs a protein structure).

**EM algorithm.** The goal of the algorithm is to find a vector K, that maps each structure to a value between 0 and 1 that maps to the biophysical property of the structure (e.g., stability). We initialize K with all 1 s for structures from variants with the biophysical property of interest, and all 0 s for structures from variants without the biophysical property of interest. Then, we alternate between M- and E-steps to update the vector K. First, the M-step fits a neural network using K as classification targets. Next, the neural network outputs a vector of scores for structures, Y. Then, we apply an E-step to update the values in K. Specifically, we compute the expected value of each structure where we treat a set of structures as binomial random variables parameterized by Y, conditioned on user-defined bounds on the number of successes (i.e., structures with the biochemical property) for each variant. The expected values are computed as the probability-weighted average of all binary realizations of binomial distributions parameterized by Y that are within the user-defined bounds. These expected values provide an updated K, allowing us to repeatedly iterate between M- and E- steps. We refer the reader to the Supplementary Information and our previous work for a more thorough discussion of the algorithm.

**Featurization.** Simulation data was preprocessed before becoming input to the DiffNets. Simulation trajectories and the original crystallographic structure (PDB ID: 1JWP) are stripped down to the XYZ coordinates of the protein backbone without carbonyl oxygens (C, CA, CB, and N). Then, the trajectories are centered at the origin and aligned to the crystallographic structure. Next, we follow a procedure similar to Wehmeyer and Noe[31] to mean-shift the XYZ coordinates to zero, followed by whitening. First, we mean shift,

$$x^{mean-free} = \sum_{i=1}^{N_t} x_i - \bar{x} \qquad (3)$$

where $x^{mean-free}$ is the mean-shifted trajectory of XYZ coordinates, $x_i$ is a single frame with XYZ coordinates, $\bar{x}$ is the mean of the XYZ coordinates across all trajectories, and $N_t$ is the number of frames in all trajectories.

Next, we whiten the data,

$$\tilde{x} = C_{00}^{-\frac{1}{2}} x^{mean-free} \qquad (4)$$

where $\tilde{x}$ is the whitened trajectory of XYZ coordinates and $C_{00}$ is the covariance matrix for the XYZ coordinates. Whitening decorrelates the inputs and adjusts their variance to unity. After whitening, we use one out of every ten simulation frames for each epoch of DiffNet training. In practice, whitening and unwhitening of the data is performed on the input XYZ coordinates directly in the DiffNet with frozen (untrainable) weights. For myosin, we subsampled the data to use only one of every ten simulation frames.

**Classification targets.** To train the model we need a target for each protein structural configuration. We assign initial, binary targets based on the observed biochemical property (e.g., 1 s for more stable variants, 0 s for less stable variants). Our assumption that individual configurations can be mapped to biochemical

properties is consistent with studies that attribute specific structural states to a biochemical property (e.g., an enzyme primed for catalysis) and designate other individual structural states as being incompatible with a biochemical property (e.g., an enzyme in an inactive state). Next, we iteratively update the initial labels with an expectation maximization algorithm (described above). This relaxes the labels such that structural configurations are on a continuum. This effectively turns the problem into a regression problem instead of a classification problem, which is consistent with the observation that most biophysical observables are on a continuum.

**Neural network training**. We trained DiffNets with three loss functions to minimize protein reconstruction error ($\ell_{Recon}$), minimize feature classification error ($\ell_{Class}$), and minimize the correlation of latent space variables ($\ell_{Corr}$).

$$\mathcal{L}_{DiffNet} = \ell_{Recon} + \ell_{Class} + \ell_{Corr} \qquad (5)$$

The reconstruction loss term attempts to tune the network weights to properly reconstruct the original XYZ coordinates of the protein. This loss combines an absolute error (L1), which funnels reconstructions to the proper XYZ coordinates, and a mean-squared error (L2) to strongly discourage outliers. Explicitly,

$$\ell_{Recon} = \frac{1}{N_b} \sum_{i=1}^{N_b} \frac{1}{N_n} \sum_{j=1}^{N_n} [|x_{ij} - \hat{x}_{ij}| + (x_{ij} - \hat{x}_{ij})^2] \qquad (6)$$

where $N_n$ is the number of output nodes (all XYZ coordinates), $N_b$ is the number of examples in a training batch, $x_{ij}$ is a target value (actual XYZ coordinate), and $\hat{x}_{ij}$ is the output value from the DiffNet.

The classification error is a binary cross entropy error that penalizes misclassifications by the latent space. This classification loss attempts to constrain the latent space to learn a dimensionality reduction that can also classify a biophysical feature. Explicitly,

$$\ell_{Class} = \frac{1}{N_b} \sum_{i=1}^{N_b} y_i * \log(\hat{y}_i) + (1 - y_i) * \log(1 - \hat{y}_i) \qquad (7)$$

where $N_b$ is the number of examples in a training batch, $y_i$ is the target value, a binary value indicating if a simulation frame has a specific feature or not, and $\hat{y}_i$ is the output of the classification layer by the DiffNet.

Finally, we include a loss function to minimize the covariance between latent space variables. This loss takes the form of

$$\ell_{Corr} = \sum_{i \neq j} Cov(z_i, z_j)^2 \qquad (8)$$

where $Cov(z_i, z_j)$ is the covariance matrix of the latent vector, Z, across all $N_b$ samples in a training batch. We reason that preventing redundancy in latent variables should maximize the amount of information one can gain in a small number of variables. Ideally, this sets us up to use just a few latent variables and still have a rich amount of information. With fewer latent variables, models are generally more interpretable.

Our training procedure uses several training iterations to progressively build in hidden layers of the DiffNet. First, we train a minimal version of a DiffNet. Explicitly, the encoders have an input layer and a reduction layer with a four-fold reduction in variables. There is no further reduction to a bottleneck layer. Instead, the decoder takes the reduction layer as input and passes it to an output layer. Training this simplified autoencoder is an easier task than training a full DiffNet because the dimensionality reduction it performs is modest. It has ~an order of magnitude more dimensions to explain the original data compared with a true bottleneck layer. We reason that this can generate useful priors for what the reduction layer should capture. In our second pretraining procedure, we freeze those priors and add the bottleneck layer in to train the full DiffNet. Therefore, this second pretraining step concentrates its representational power on tuning how to properly reduce from the reduction layer to the bottleneck layer. Finally, we unfreeze the priors and train the full DiffNet to polish all weights. Each of these three procedures undergoes 20 training epochs. In the self-supervised setting, classification labels are updated using expectation maximization after each training epoch.

All training was performed in PyTorch 1.1[55]. Training on ~120,000 simulation frames of β-lactamase takes under one hour on a single AMD Vega 20 GPU. Training with expectation maximization approximately doubled the training time for DiffNets trained on TEM. We used the Adam optimizer with a learning rate of 0.0001 and a batch size of 32.

We performed limited hyperparameter tuning to arrive at our final models. We found that the DiffNet performance was robust across a wide range of latent variables (Fig. 3) and expectation maximization bounds (Supplementary Fig. 1, Fig. 5b). To choose a final number of latent variables, we chose the minimum number where reconstruction error no longer showed qualitative improvement. Additionally, we saved a trained model after every epoch of training and ultimately used the model that showed the best reconstruction performance on a validation set that contained 10% of the data.

**Reconstruction experiment**. To analyze DiffNet reconstruction error (Fig. 3), we trained on five architectures where we varied the numbers of latent variables. All architectures split the input (as in Fig. 1b) such that any atom (C, CA, N, CB) within 1 nm of residue 182 (source of single point mutation – colored cyan in Fig. 6) was included in encoder A, while the rest of the protein was included in encoder B. Encoder A reduced down to 1, 2, 3, 5, and 10 latent variables, while encoder B reduced down to 2, 3, 7, 20, and 40 latent variables. After training, we use the neural networks to reconstruct the protein structure from 1 of every 100 simulation frames and compute its root-mean squared deviation from the actual structure obtained via simulation.

**Classification labels**. To provide classification labels for Fig. 4, we designated simulation frames as "compact helix" or "extended helix" based on a previous manuscript that identified three key hydrogen bond distances in Helix 9 that distinguish stabilizing variants from nonstabilizing variants (Res 182-186, Res 183-187, and Res 186-190)[5]. Specifically, we label helix 9 compact if the distance between the backbone nitrogen and the carbonyl oxygen is less than 4.2 Ångstroms for all residue pairs listed, and we label it extended otherwise.

**β-lactamase expectation maximization experiment**. When training on β-lactamase with expectation maximization (Figs. 5 and 6) we trained a split architecture DiffNet consisting of 2 encoders and 2 latent spaces (as visualized in Fig. 1). The input to encoder "A" is all XYZ coordinates within 1 nm of residue 182 (1 nm region around the mutation). The input to encoder "B" is the XYZ coordinates from the rest of the protein. These encoders reduce the input to 4 and 26 latent variables, respectively (30 total latent variables split proportionally into latent A and latent B based on the number of atoms input into each encoder). After training, we applied the trained DiffNet to all simulation data to obtain DiffNet output labels. These output labels can be thought of as a proxy for latent A (region around the mutation) as the output label is simply a linear combination of the values in latent A (then scaled between 0 and 1 using the PyTorch sigmoid activation function). We bin all structures into 10 equally spaced bins from 0-1 based on their DiffNet output label. Then, we measure the average distance for Res 182-186, Res 183-187, and Res 186-190 in each bin (Fig. 5a). To find distance changes that are correlated with changes in the DiffNet output label (as shown in Fig. 6), we first cluster the simulation data into 2000 clusters using a hybrid k-centers and k-medoids approach with our open-source python package, Enspara[56]. Then, we enumerate all possible distance pairs between atoms in encoder A (i.e., within 1 nm of the mutation). For each distance pair, we perform a linear regression between the distance and the DiffNet output label across all 2000 cluster centers. We then select the distance pairs with the highest correlation coefficients (top 1%) and visualize them in PyMol (Fig. 6).

**Myosin expectation maximization experiment**. When training on myosin (Fig. 7) we used an architecture with a single encoder (i.e., not split) that received C, CA, N, and CB atoms as input within a 2.25 nm radius around the P-loop (specifically residue S180, *Myh7* numbering). We used 50 latent variables. All frames from low duty motors were initially assigned classification labels of 0, and simulation frames from high duty motors were initially assigned 1 s. During the EM procedure, we set bounds of 10–40% for low duty motor frames and 60–90% for high duty motor frames. To find distance changes that are correlated with changes in the DiffNet output label, we copied the scheme described in the previous section. To identify P-loop orientations with high/low DiffNet labels, we selected the 10 structures with DiffNet labels closest to 0.03 and 0.7 from a single isoform (Myh7). To calculate the cumulative distribution function in Fig. 7e, we used a previously published MSM. Specifically, for each cluster center in the MSM, we measured its distance and weighted the distance by its equilibrium population[47]. Lastly, for Fig. 7c, we grouped structures as nucleotide favorable/unfavorable as defined in a previous manuscript[47].

**Reporting summary**. Further information on research design is available in the Nature Research Reporting Summary linked to this article.

## Data availability
The datasets are not publicly deposited because they are several terabytes in size. The datasets generated during and/or analyzed during the current study are available from the corresponding author on reasonable request. We expect that it should take several business days to share the data upon a particular request. Once shared, we will not enforce any limitations for how the data may be used.

## Code availability
Data normalization and DiffNets training with, or without, expectation maximization is freely available on GitHub at https://github.com/bowman-lab/diffnets.

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

## Acknowledgements

This work was funded by NSF CAREER Award MCB-1552471 and NIH grants R01 GM124007 and RF1AG067194 (G.R.B.). We would like to thank AMD for the donation of critical hardware and support resources from its HPC Fund that enabled the computations for this work. G.R.B. holds a Career Award at the Scientific Interface from the Burroughs Wellcome Fund and a Packard Fellowship for Science and Engineering from The David & Lucile Packard Foundation. M.D.W. was supported by a MolSSI COVID-19 seed software fellowship and would like to thank Sina Mostafanejad and Doaa Altarawy for their guidance in developing DiffNets as a software package.

## Author contributions

M.D.W., M.I.Z., A.M., M.C., S.J.S. and G.R.B. conceptualized different aspects of the research and developed software. M.D.W. and G.R.B. acquired funding. M.D.W., M.I.Z., A.M., M.C. and G.R.B. conducted research. M.D.W. wrote the original draft and all other authors reviewed and edited the manuscript. M.D.W. M.I.Z., A.M., and M.C. created the data visualizations.

## Competing interests

The authors declare no competing interests.
