## [Peer Review File · Nature Communications]

Reviewers' Comments:

Reviewer #1:

Remarks to the Author:

This paper describes the development and application of DiffNet, an approach to mathematically encode protein structures in a manner that enables easy mapping to functional properties of interest. The way they achieve this is training an autoencoder augmented with a classification task to learn structural features that are predictive of biochemical properties such as thermostability. The innovations of this paper are twofold:

- The addition of a classification task to an autoencoder to help the model learn a representation that reflects the difference between the two biochemical states
- An expectation-maximization algorithm to label individual structures in structural ensembles with biological properties

These innovations are demonstrated in three studies:

1) Predicting whether a variant of TEM β -lactamase has helix 9 compaction, a feature indicative of stability. The authors found that adding a classification task to the model allows it to learn latent representations that reflect this property.

2) Predicting whether a variant of TEM β -lactamase is stable. The authors sought to predict a biochemical property (stability) for individual structures within an ensemble. They applied expectation maximization to iteratively update the stability label of individual structures in the structure ensemble of each TEM β -lactamase variant and showed that the model predictions recapitulate the correlation between stability and helix 9 compaction. They also showed that the model identifies other structural features important for stability.

3) Generalizability of DiffNet. The authors applied DiffNet to a new variant of TEM β -lactamase and showed that the model can predict the stability of individual structures in its ensemble. They also applied the model to classify other protein families (high duty myosin) and demonstrated that it extracts biochemically-sensible structural features.

The paper is generally well-written. It presents an algorithm that can extract biochemically meaningful features from MD simulations to predict properties of interest. While the addition of a classification task to an autoencoder is not really a methodological innovation, its combination with their expectation-maximization scheme is. Taken together these changes are sound and are of interest to 1) scientists who use molecular dynamics to study conformation ensembles and understand the biochemical properties of a protein and 2) a wider community who are interested in model interpretability and biophysicality. A significant aspect of their result is that they are able to capture functional differences even when the underlying structural changes are minor, which is a challenging task.

Below are some specific comments.

- The authors demonstrated that their approach automatically extracts structural features predictive of stability. While they showed that the tandem classification task helped in learning a predictive latent representation, the effect of EM on feature extraction remains unclear. Specifically, what would the results in Figure 5 and Figure 6 look like if the labels are fixed during training? This comparison will be critical for dissecting whether the biochemically intuitive approach of allowing individual structures in the structure ensemble to have different properties actually leads to better model interpretability.
- How is the constraint value (0-30% for the less stable variants and 60-90% for the more stable variants) determined? How sensitive are the models to these values? Do these values affect results in

Fig 7?

- Is it possible to assess whether the prediction of the individual structures are correct?

- Some of the results were stated without an illustrative figure and it is difficult to picture the structural features described. For example, show inset with the specific mutation mentioned in "the distance correlations show a compaction between residue 189, in the middle of helix 9, and residue 263 on the β -sheet, which both have chemical shift perturbations for M182T and M182S (more stable variants) relative to the wild-type." Label the specific mutations. Similarly, adding a figure to illustrate "In particular, there are two residues on switch-II with distances that are strongly correlated with the DiffNet label indicating that conformational changes in switch-II may be important for determining the duty ratio" would also help.

Reviewer #2:

Remarks to the Author:

In their paper, the authors present a novel deep learning model for identifying zones in proteins which affect, and therefore are predictive of, biochemical characters in protein variants. Septically, the authors develop a self-supervised version of autoencoder, that allows the model to learn a low-dimensional representation of proteins (like in normal autoencoders), yet while also conserving the ability to differ between them according to specific properties. The capability of the model is demonstrated by two predictive tasks: one helps to identify structural differences that are predictive of the stability of different variants of beta-lactamase; and another one learns the differences which predictive of duty ratio in myosin. This is overall an interesting and worthy of publication study.

Yet, I do have some comments/concerns:

1) The paper is not well positioned within the literature. It is nor described in the intro what have been done before, what is the novelty in this research. In particular, how does the work compare with the recent advancement of AlphaFold? There are also several related publications on protein autoencoders from Debora Marks' group ('Deep generative models of genetic variation capture the effects of mutations' for an example).

2) Figure 5 seems like a mathematically trivial (a tautology): if you train a model supervised by stability, and you also know that stability is correlated with helix 9 compaction, it mathematically follows from transitivity of correlations that the model will succeed to show a correlation between the compaction and the stability.

3) It's unclear from the text how the authors have reconstructed or represent the physical structure of the protein from the latent layer, i.e. how do you know to tell which zones in the protein's structure are correlated with specific features in the latent layer.

Reviewer #3:

Remarks to the Author:

Review of "DiffNets: deep learning the structural determinants of proteins biochemical properties by comparing different structural ensembles" by Ward, Zimmerman, Meller, Chung, Swamidass, and Bowman.

In their manuscript, Ward and colleagues propose a deep learning approach to learn latent space representations of protein conformations that correlate with macroscopic biochemical measurements (properties). Such a method could potentially be helpful when analysing high-dimensional molecular

simulation data when experimental data is available, and a structural interpretation is sought.

The authors describe a multi-task auto-encoder inspired architecture which they call DiffNet. Briefly, they link a conventional auto-encoder to a classification task, where the learned latent code is used for binary classification: for example high or low stability etc. In their simplest form DiffNets are trained on simulation data from the two different classes where the conformations are mapped to a label given by the biochemical observable. As the authors note, this may be limiting since there may be overlap in the conformational ensembles of the molecular systems with different properties. Consequently, they introduce an "expectation-maximization" algorithm to relax label assignments within some bounds, in a "self-supervised" way. In this way, the fraction of frames a constrained into a certain range depending on the system they come from: high or low property measure.

Overall, it appears that DiffNet indeed learns latent space embeddings which are able to link structural features to a macroscopic observable. The authors illustrate this nicely with their application a number of simulations of beta-lactamases variants. The Bowman lab are experts in the computational and biophysical characterization of beta-lactamases and yet DiffNets are able to detect conformational signatures which they say took them months to identify using conventional strategies.

Other results, such as those presented in the section "DiffNets predictive power extends to variants not seen during training." are much less convincing. The authors show highly skewed prediction distributions on the data used for training, whereas the validation dataset, display the probability density in between the classes. To this readers interpretation, this means that it will assign the majority of the simulation frames to a molecular property at random. The authors argue that there are peaks close to zero and one, yet one of them is dominating -- yet, classifications based on distributions like this will have substantial uncertainties. Nevertheless, the authors make no note of this potential problem. This reader, is not convinced that DiffNets, in their current implementation can generalize well beyond the training set.

Fundamentally, the approach makes a number of assumptions which may be limiting:

- Individual configurations are assumed to maps to an observable value (albeit, binarized), although many of these observables are only meaningfully defined for the whole thermodynamic ensemble.
- Observables are either low or high in the value. However, mostly biophysical observables are continuous values.

While DiffNets may have their limitations, they appear to be a useful tool for analysing molecular dynamics where some qualitative experimental observables are available. Indeed, the authors do show several interesting results to this end. Consequently, it may be a first of many such developments, where the stark and possibly limiting assumptions are relaxed in a rigorous theoretical framework. Nevertheless, this reader does not expect DiffNets to neither make quantitative predictions nor generalize beyond their training data. The authors are therefore strongly advised to tone down their conclusions and align their conclusions with the results and limitations of the presented method.

Minor comments:

In figure 5. error-bars are missing.

Expectation minimization algorithm is discussed in qualitative and vague manner. The authors are encouraged to provide a more in-depth discussion of the algorithm and state clearly assumptions made, for example in a supplement.

REVIEWER COMMENTS

Reviewer #1 (Expertise: ML for predicting protein function from structure/sequence):

This paper describes the development and application of DiffNet, an approach to mathematically encode protein structures in a manner that enables easy mapping to functional properties of interest. The way they achieve this is training an autoencoder augmented with a classification task to learn structural features that are predictive of biochemical properties such as thermostability. The innovations of this paper are twofold:

- The addition of a classification task to an autoencoder to help the model learn a representation that reflects the difference between the two biochemical states
- An expectation-maximization algorithm to label individual structures in structural ensembles with biological properties

These innovations are demonstrated in three studies:

- 1) Predicting whether a variant of TEM 13-lactamase has helix 9 compaction, a feature indicative of stability. The authors found that adding a classification task to the model allows it to learn latent representations that reflect this property.
- 2) Predicting whether a variant of TEM 13-lactamase is stable. The authors sought to predict a biochemical property (stability) for individual structures within an ensemble. They applied expectation maximization to iteratively update the stability label of individual structures in the structure ensemble

Campus Box 8231, 660 South Euclid Avenue, St. Louis, Missouri 63110 <http://bowmanlab.biochem.wustl.edu/>
Direct TEL: (314) 362-7433 Direct FAX: (314) 362-7183 [e-mail: bowman@biochem.wustl.edu](mailto:bowman@biochem.wustl.edu)

of each TEM 13-lactamase variant and showed that the model predictions recapitulate the correlation between stability and helix 9 compaction. They also showed that the model identifies other structural features important for stability.

3) Generalizability of DiffNet. The authors applied DiffNet to a new variant of TEM 13-lactamase and showed that the model can predict the stability of individual structures in its ensemble. They also applied the model to classify other protein families (high duty myosin) and demonstrated that it extracts biochemically-sensible structural features.

The paper is generally well-written. It presents an algorithm that can extract biochemically meaningful features from MD simulations to predict properties of interest. While the addition of a classification task to an autoencoder is not really a methodological innovation, its combination with their expectation-maximization scheme is. Taken together these changes are sound and are of interest to 1) scientists who use molecular dynamics to study conformation ensembles and understand the biochemical properties of a protein and 2) a wider community who are interested in model interpretability and biophysicality. A significant aspect of their result is that they are able to capture functional differences even when the underlying structural changes are minor, which is a challenging task.

--Thank you for these nice comments!

Below are some specific comments.

- The authors demonstrated that their approach automatically extracts structural features predictive of stability. While they showed that the tandem classification task helped in learning a predictive latent representation, the effect of EM on feature extraction remains unclear. Specifically, what would the results in Figure 5 and Figure 6 look like if the labels are fixed during training? This comparison will be critical for dissecting whether the biochemically intuitive approach of allowing individual structures in the structure ensemble to have different properties actually leads to better model interpretability.

--Good point, thank you. We have compared a supervised autoencoder with and without EM and reproduced Figure 5 with data from both. This result shows that EM does indeed improve the autoencoders ability to label structures based on their association with stability. To better demonstrate this we have updated the figure and the text:

“Expectation maximization aids the DiffNet in learning a low-dimensional representation that accurately identifies that helix 9 compaction is associated with highly stable variants. First, we trained two supervised autoencoders (one with and one without expectation maximization) and compared the distribution of output classification labels. Without expectation maximization, almost all structures from more stable variants have output labels close to 1, and structures from less stable variants have output labels close to 0 (Fig. 5a). This is at odds with the fact that there is structural overlap between the ensembles. It indicates that the supervised autoencoder essentially memorizes which ensemble each structure comes from instead of learning a useful association between individual structures and stability. In contrast, when expectation maximization is applied the output labels span the full spectrum from 0-1 for each variant (Fig. 5b), which is consistent across a wide range of expectation maximization bounds (Supp. Fig. 1). Moreover, as the labels increase from 0 to 1, helix 9 compaction smoothly decreases, which indicates that DiffNets learn a latent space with a continuum of structures that are less/more closely associated with stability (Fig. 5c). Without expectation maximization, the extreme labels (i.e. 0,1) track well with helix stability, but structures labelled between 0.1 and 0.9 do not show a clear trend of helix compaction.”

Figure 5: Self-supervision improves the DiffNet's ability to organize structural configurations based on their biochemical property. Histogram showing DiffNet output labels across all simulation frames from M182T and M182S (red – highly stable variants in training set) versus WT and M182V (grey – less stable variants in training set) for a supervised autoencoder (a) and a self-supervised autoencoder (b). (c) Three key hydrogen bond lengths in helix 9 as a function of the DiffNet output label (yellow – supervised, black – self-supervised), which ranges from zero for structures associated with low stability to one for structures associated with high stability. The distances are between the carbonyl carbon of the i 'th residue and the nitrogen of the $(i+4)$ 'th residue. Standard error bars are not visible since the standard error is smaller than scatter points.

--Second, we compared the structural changes suggested by DiffNets with and without EM for for both 13-lactamase and the myosin motors. Because 13-lactamase is a relatively rigid protein, we only expect to see subtle differences. In contrast, we expect more transformative improvements when applying EM to the more flexible myosin motors. Indeed, Supplementary

Figure 3 shows that models trained with and without EM give highly similar results for 13-lactamase. However, we observe a dramatic difference for myosin. Specifically, changes in Ser180 structural states were previously identified as the most important indicator for determining duty ratio of myosin proteins. This is strongly detected when EM is applied, but without EM it is not detected at all. We include both comparisons in a supplementary figure and update the text as follows:

“This analysis finds 78 distances between the P-loop and the loop connecting a nearby loop involved in ADP handling (referred to as the 133-SH2 loop), again highlighting that the DiffNet learns that P-loop dynamics are important for discriminating high and low duty motors. **We compared this result to a model trained without expectation maximization and find that expectation maximization improves the quality of this analysis. Specifically, changes in Ser180 are strongly detected when expectation maximization is applied, but without expectation maximization these changes are not detected at all. (Supp. Fig. 3).**”

Figure S3. Impact of expectation maximization on what features a DiffNet uses to distinguish variants. Dotted lines indicate distances between two atoms that change in a way that is strongly correlated with an increased DiffNet output label. Red indicates the atoms move closer together as the output label increases, blue indicates atoms moving away from each other. Results for 13-lactamase variants and myosin are shown in **(a)** and **(b)** respectively. In **(a)**, protein atoms are colored cyan if they are near the mutation, which indicates that they were

included in the classification task and considered for the distance correlation calculation. The site of the single point mutation is highlighted with a yellow sphere.

- How is the constraint value (0-30% for the less stable variants and 60-90% for the more stable variants) determined? How sensitive are the models to these values? Do these values affect results in Fig 7?

--The constraint values were originally selected based on a qualitative intuition to allow some overlap, but still have a clear signal to distinguish variants. We agree that further rigor is necessary here. We have tested a set of bounds that allow progressively more overlap (0-40% and 50-90%, 0-50% and 40-90%, 0-60% and 30-90%, 0-70% and 20-90%) and have included a supplementary figure to show the effect of modulating these constraints. For each set of bounds, we plot the data as we did in Figure 5 and Figure 7 (Supp. Figure 1). We find that a DiffNet's ability to learn a latent space with a continuum of structures that are less/more associated with stability is not very sensitive to the specific choice of bounds (Supp. Fig. 1b). We also find that the results in Fig. 7 are mostly consistent across EM bounds.

“During the EM procedure, we calculated the expected values (updated labels) conditioned on the constraint that 0-30% of less stable variants frame are likely to be stabilizing, and 60-90% of frames for highly stable variants. In general, it should be sufficient to base bounds on qualitative *a priori* knowledge rather than precise, quantitative information. In this case, we chose these bounds as a way to allow overlap between ensembles, but still provide a clear signal to distinguish more and less stable variants. Empirically, we find that DiffNets are robust across a wide range of bounds (Supp. Fig. 1).”

“...In contrast, when expectation maximization is applied the output labels span the full spectrum from 0-1 for each variant (Fig. 5b), which is consistent across a wide range of expectation maximization bounds (Supp. Fig. 1). Moreover, across all structures, as the labels increase from 0 to 1, helix 9 compaction smoothly decreases, which indicates that DiffNets learn a latent space with a continuum of structures that are less/more closely associated with stability (Fig. 5c).”

Figure S1. Self-supervised DiffNets are robust across a range of expectation maximization bounds. (a) Histogram showing DiffNet output labels across all simulation frames from M182T and M182S (red – highly stable variants in training set) versus WT and M182V (grey – less stable variants in training set) across a range of expectation maximization bounds. Predictions on a less stable variant not seen during training (M182N) are also shown (black dotted line). (b) Three key hydrogen bond lengths in helix 9 as a function of the DiffNet output label (yellow – supervised, black – self-supervised), which ranges from zero for structures associated with low stability to one for structures associated with high stability. The distances are between the carbonyl carbon of the i 'th residue and the nitrogen of the $(i+4)$ 'th residue. Standard error bars are not visible since the standard error is smaller than scatter points.

- Is it possible to assess whether the prediction of the individual structures are correct?

--We believe that the updated figure 5 demonstrates that predictions on individual structures are mostly correct. Given that it is impossible to experimentally measure the stability of individual structures, I don't think we can address this further.

- Some of the results were stated without an illustrative figure and it is difficult to picture the structural features described. For example, show inset with the specific mutation mentioned in "the distance correlations show a compaction between residue 189, in the middle of helix 9, and residue 263 on the β -sheet, which both have chemical shift perturbations for M182T and M182S (more stable variants) relative to the wild-type." Label the specific mutations. Similarly, adding a figure to illustrate "In particular, there are two residues on switch-II with distances that are strongly correlated with the DiffNet label indicating that conformational changes in switch-II may be important for determining the duty ratio" would also help.

--Thanks for pointing this out! We have updated figure 6 to highlight the mutated residue with a yellow sphere. We have also added an inset that better highlights the comparison to NMR data. Specifically, we visually highlight residues that have chemical shift perturbations in the most stable variant (M182S) relative to the wild-type, instead of mentioning specific residues. We adjust the text accordingly:

"For example, our NMR data identified that the β -sheet adjacent to helix 9 differs in more stable vs less stable variants,⁵ but our computational analysis did not detect a clear trend. On the same simulation dataset, the DiffNet does detect differences in the β -sheet between more and less stable variants. **Specifically, the DiffNet analysis suggests more stable variants have tighter packing at the helix 9 and β -sheet interface**

(Fig. 6b). Often times the important features that distinguish protein variants can be complicated and, therefore, easily missed even with months of analysis. DiffNets can learn complicated features and help automate the process of identifying biochemically relevant structural features that distinguish protein variants.”

Figure 6: Visualization of the features that DiffNets find important for increased stability of M182T and M182S variants. (a) Dotted lines indicate distances between two atoms that change in a way that is strongly correlated with an increased DiffNet output label. Red indicates the atoms move closer together as the output label increases, blue indicates atoms moving away from each other. The mutated residue is highlighted with a yellow sphere. Protein atoms are colored cyan if they are near the mutation, which indicates that they were included in the classification task and considered for the distance correlation calculation. (b) Rotated inset of (a) showing DiffNet predicted packing at the interface of helix 9 and the adjacent β -sheet. Residues with chemical-shift perturbations in M182S relative to wild-type are shown in deep olive.

--We also add a figure and short paragraph in the supplement to better illustrate the DiffNet predicted changes in switch II.

Figure S4. DiffNet analysis suggests conformational changes on switch-II are important for distinguishing high-and low-duty myosin isoforms. Dotted lines indicate distances between two atoms that change in a way that is strongly correlated with an increased DiffNet output label. Red indicates the atoms move closer together as the output label increases, blue indicates atoms moving away from each other. Switch-II is colored orange and the p-loop is colored purple.

“Self-supervised DiffNet predicts that distance changes involving residues on switch-II (F468, E466) distinguish high and low-duty motor myosins. A series of myosin-6 crystal structures¹ suggests that motions in switch-II are required to open a phosphate release pathway. Moreover, E466 directly coordinates phosphate at the end of this putative phosphate release tunnel¹. Thus, DiffNets may be identifying important motions required for phosphate release, a key determinant of duty ratio.”

1. Llinas P, Isabet T, Song L, et al. How Actin Initiates the Motor Activity of Myosin. *Dev Cell*. 2015. doi:10.1016/j.devcel.2015.03.025

Reviewer #2 (Expertise: Experimental B-lactamase sequence-to-function characterization):

In their paper, the authors present a novel deep learning model for identifying zones in proteins which

affect, and therefore are predictive of, biochemical characters in protein variants. Septically, the authors develop a self-supervised version of autoencoder, that allows the model to learn a low-dimensional representation of proteins (like in normal autoencoders), yet while also conserving the ability to differ between them according to specific properties. The capability of the model is demonstrated by two predictive tasks: one helps to identify structural differences that are predictive of the stability of different variants of beta-lactamase; and another one learns the differences which predictive of duty ratio in myosin. This is overall an interesting and worthy of publication study.

--Thank you for the words of affirmation!

Yet, I do have some comments/concerns:

1) The paper is not well positioned within the literature. It is nor described in the intro what have been done before, what is the novelty in this research. In particular, how does the work compare with the recent advancement of AlphaFold? There are also several related publications on protein autoencoders from Debora Marks' group ('Deep generative models of genetic variation capture the effects of mutations' for an example).

--Thank you for the feedback. The recent advancement of AlphaFold allows one to predict the native folded state of proteins, but as mentioned in the introduction, comparing native folded states of protein variants is insufficient to explain their biochemical differences. DiffNets is a dimensionality reduction algorithm used to compare structural ensembles to identify biochemically relevant structural differences, which is a task that is mostly unrelated to AlphaFold. In the introduction, we discuss other dimensionality reduction algorithms that are commonly applied to molecular dynamics simulations (e.g. autoencoders) and then explain the novelty in our approach. Namely, a DiffNet is similar to a standard autoencoder but it is augmented with a classification task and an expectation maximization algorithm to learn a low-dimensional representation that makes it easier to find differences between protein variant structural ensembles.

The work you mentioned from Debora Marks' group is an excellent piece of work. There is some overlap between the works, but the core goals of each approach are quite different. Our model learns a latent representation of data from computer simulations of proteins that helps identify a structural mechanism that explains biochemical differences between protein variants. Their work learns a latent representation of data from a protein sequence that can predict mutational effects (but has no explanatory power with regard to structural mechanisms). The section with the most overlap with their work is figure 7 where we correctly predict the effect of the M182N mutation. Based on your point we do include a reference to their work in the appropriate section.

“Using DiffNets to predict on a variant outside of training provides further support that expectation maximization aids in learning structural features associated with stability. We compared each model’s ability to predict the stability of a less stable variant not seen during training (M182N), and we find that this prediction is improved when expectation maximization is applied (Supp. Fig. 2). This suggests expectation maximization helps DiffNets hone in on biochemically relevant structural features, and that DiffNets could be used as a predictive tool. However, we caution that autoencoders will fail anytime they are applied to data that is highly dissimilar from the training set, so a DiffNet will not perform well on new variants that visit conformations not visited in the training set. Future studies would be necessary to optimize DiffNets for prediction and should be evaluated against related methods such as by Riesselman et al.”

2) Figure 5 seems like a mathematically trivial (a tautology): if you train a model supervised by stability, and you also know that stability is correlated with helix 9 compaction, it mathematically follows from transitivity of correlations that the model will succeed to show a correlation between the compaction and the stability.

--Thank you for raising this interesting point, you are certainly correct. One goal is to show that the low output labels have extended helices (i.e. less stable) and high output labels have compact helices (i.e. more stable). As you mention, this is mathematically trivial / expected – but it first provides a sanity check that the classification task (and expectation maximization update to the labels) is not having unexpected/poor behavior. The more interesting goal is to show that not only are the lowest and highest output labels correct, but that the latent space has a continuum of structures from 0 to 1 that are less/more closely related to stability. To demonstrate this point, we added another model to the original figure where the model is supervised by stability with fixed labels (i.e. no expectation maximization). For this model, though the endpoints are correct, the intermediate values do not show a clear trend. However, with our expectation maximization scheme, there is a clear gradient of compaction as output labels increase from 0 to 1, which indicates that structures are organized in the latent space based on if they are less/more closely associated with stability. We’ve updated the main text as follows:

“Expectation maximization aids the DiffNet in learning a low-dimensional representation that accurately identifies that helix 9 compaction is associated with highly stable variants. First, we trained two supervised autoencoders (one with and one without expectation maximization) and compared the distribution of output classification labels. Without expectation maximization, almost all structures from more stable variants have output labels close to 1, and structures from less stable variants have output labels close to 0 (Fig. 5a). This is at odds with the fact that there is structural overlap between the ensembles. It indicates that the supervised autoencoder essentially memorizes which ensemble each structure comes from instead of learning a useful association between individual structures and stability. In contrast, when expectation maximization is applied the output labels span the full spectrum from 0-1 for each variant (Fig. 5b), which is consistent across a wide range of expectation maximization bounds (Supp. Fig. 1). Moreover, as the labels increase from 0 to 1, helix 9 compaction smoothly decreases, which indicates that DiffNets learn a latent space with a continuum of structures that are less/more closely associated with stability (Fig. 5c). Without expectation maximization, the extreme labels (i.e. 0,1) track well with helix stability, but structures labelled between 0.1 and 0.9 do not show a clear trend of helix compaction.”

Figure 5: Self-supervision improves the DiffNet's ability to organize structural configurations based on their biochemical property. Histogram showing DiffNet output labels across all simulation frames from M182T and M182S (red – highly stable variants in training set) versus WT and M182V (grey – less stable variants in training set) for a supervised autoencoder (**a**) and a self-supervised autoencoder (**b**). (**c**) Three key hydrogen bond lengths in helix 9 as a function of the DiffNet output label (yellow – supervised, black – self-supervised), which ranges from zero for structures associated with low stability to one for structures associated with high stability. The distances are between the carbonyl carbon of the i 'th residue and the nitrogen of the $(i+4)$ 'th residue. Standard error bars are not visible since the standard error is smaller than scatter points.

3) It's unclear from the text how the authors have reconstructed or represent the physical structure of

the protein from the latent layer, i.e. how do you know to tell which zones in the protein's structure are correlated with specific features in the latent layer.

--Thanks for raising this point, hopefully I can address the confusion by updating the Methods:

"When training on β -lactamase with expectation maximization (Fig 5, 6, 7) we trained a split architecture DiffNet consisting of 2 encoders and 2 latent spaces (as visualized in Fig. 1). The input to encoder "A" is all XYZ coordinates within 1nm of residue 182 (1nm region around the mutation). The input to encoder "B" is the XYZ coordinates from the rest of the protein. These encoders reduce the input to 4 and 26 latent variables, respectively (30 total latent variables split proportionally into latent A and latent B based on the number of atoms input into each encoder). After training, we applied the trained DiffNet to all simulation data to obtain DiffNet output labels. These output labels can be thought of as a proxy for latent A (region around the mutation) as the output label is simply a linear combination of the values in latent A (then scaled between 0 and 1 using the PyTorch sigmoid activation function). We bin all structures from the simulation data into 10 equally spaced bins from 0-1 based on the DiffNet output label. Then, we measure the average distance for Res 182-186, Res 183-187, and Res 186-190 in each bin (Figure 5c). Next, to find distance changes that are correlated with changes in the DiffNet output label (as shown in Figure 6), we first cluster the simulation data into 2000 clusters using a hybrid k-centers and k-medoids approach with our open-source python package, Enspara⁵⁴. Then, we enumerate all possible distance pairs between atoms in encoder A (i.e. within 1nm of the mutation). For each distance pair, we perform a linear regression between the distance and the DiffNet output label across all 2000 cluster centers. We then select the distance pairs with the highest correlation coefficients (top 1%) and visualize them in PyMol (Figure 6)."

Reviewer #3 (Expertise: ML for predicting protein function from structure/sequence):

Review of "DiffNets: deep learning the structural determinants of proteins biochemical properties by comparing different structural ensembles" by Ward, Zimmerman, Meller, Chung, Swamidass, and Bowman.

In their manuscript, Ward and colleagues propose a deep learning approach to learn latent space representations of protein conformations that correlate with macroscopic biochemical measurements (properties). Such a method could potentially be helpful when analysing high-dimensional molecular simulation data when experimental data is available, and a structural interpretation is sought.

The authors describe a multi-task auto-encoder inspired architecture which they call DiffNet. Briefly, they link a conventional auto-encoder to a classification task, where the learned latent code is used for binary classification: for example high or low stability etc. In their simplest form DiffNets are trained on simulation data from the two different classes where the conformations are mapped to a label given by the biochemical observable. As the authors note, this may be limiting since there may be overlap in the conformational ensembles of the molecular systems with different properties. Consequently, they introduce an "expectation-maximization" algorithm to relax label assignments within some bounds, in a "self-supervised" way. In this way, the fraction of frames a constrained into a certain range depending on the system they come from: high or low property measure.

Overall, it appears that DiffNet indeed learns latent space embeddings which are able to link structural features to a macroscopic observable. The authors illustrate this nicely with their application a number of simulations of beta-lactamases variants. The Bowman lab are experts in the computational and

biophysical characterization of beta-lactamases and yet DiffNets are able to detect conformational signatures which they say took them months to identify using conventional strategies.

--Thanks for the kind words!

- Other results, such as those presented in the section "DiffNets predictive power extends to variants not seen during training." are much less convincing. The authors show highly skewed prediction distributions on the data used for training, whereas the validation dataset, display the probability density in between the classes. To this readers interpretation, this means that it will assign the majority of the simulation frames to a molecular property at random. The authors argue that there are peaks close to zero and one, yet one of them is dominating -- yet, classifications based on distributions like this will have substantial uncertainties. Nevertheless, the authors make no note of this potential problem. This reader, is not convinced that DiffNets, in their current implementation can generalize well beyond the training set.

While DiffNets may have their limitations, they appear to be a useful tool for analysing molecular dynamics where some qualitative experimental observables are available. Indeed, the authors do show several interesting results to this end. Consequently, it may be a first of many such developments, where the stark and possibly limiting assumptions are relaxed in a rigorous theoretical framework. Nevertheless, this reader does not expect DiffNets to neither make quantitative predictions nor generalize beyond their training data. The authors are therefore strongly advised to tone down their conclusions and align their conclusions with the results and limitations of the presented method.

--Thank you for your feedback. As you mentioned above, the core value of DiffNets is in their ability to provide insight into structural differences between variants with biochemical properties. We have removed the section "**DiffNets predictive power extends to variants not seen during training**" and reduced the size and importance of the findings in the manuscript. Specifically, we move figure 7 to the supplement. And, instead of emphasizing its predictive power, we instead argue that the "correct" prediction relative to the incorrect prediction made by a supervised autoencoder (without expectation maximization) adds support that expectation maximization aids our method in learning general features of stability. The updated text is as follows:

"Using DiffNets to predict on a variant outside of training provides further support that expectation maximization aids in learning structural features associated with stability. We compared each model's ability to predict the stability of a less stable variant not seen during training (M182N), and we find that this prediction is improved when expectation maximization is applied (Supp. Fig. 2). This suggests expectation maximization helps DiffNets hone in on biochemically relevant structural features, and that DiffNets could be used as a predictive tool. However, we caution that autoencoders will fail anytime they are applied to data that is highly dissimilar from the training set, so a DiffNet will not perform well on new variants that visit conformations not visited in the training set. Future studies would be necessary to optimize DiffNets for prediction and should be evaluated against related methods such as by Riesselman et al."

Figure S2: Self-supervised DiffNets improve ability to predict properties of variants outside the training. Histogram of final DiffNet output labels for all simulation data points organized by variant (red – more stable variants, grey – less stable variants, black – less stable variant not seen during training) for a self-supervised DiffNet and a supervised DiffNet.

- Fundamentally, the approach makes a number of assumptions which may be limiting:
- Individual configurations are assumed to map to an observable value (albeit, binarized), although many of these observables are only meaningfully defined for the whole thermodynamic ensemble.
- Observables are either low or high in the value. However, mostly biophysical observables are continuous values.

--Thanks for the feedback, we have updated the methods to address both of these comments.

Classification Targets

To train the model we need a target for each protein structural configuration. We assign initial, binary targets based on the observed biochemical property (e.g. 1s for more stable variants, 0s for less stable variants). Our assumption that individual configurations can be mapped to biochemical properties is consistent with studies that attribute specific structural states to a biochemical property (e.g. an enzyme primed for catalysis) and designate other individual structural states as being incompatible with a biochemical property (e.g. an enzyme in an inactive state). Next, we iteratively update the initial labels with an expectation maximization algorithm (described above). This relaxes the labels such that structural configurations are on a continuum. This effectively turns the problem into a regression problem instead of a classification problem, which is consistent with the observation that most biophysical observables are on a continuum.

Minor comments:

In figure 5. error-bars are missing.

--We have the standard error bars plotted. The bars are very small because of the large amount of data.

Expectation minimization algorithm is discussed in qualitative and vague manner. The authors are encouraged to provide a more in-depth discussion of the algorithm and state clearly assumptions made, for example in a supplement.

--Thank you, we hope the following passage in the Methods clears up your concerns.

“EM algorithm

The goal of the algorithm is to find a vector K , that maps each structure to a value between 0 and 1 that maps to the biophysical property of the structure (e.g. stability). We initialize K with all 1s for structures from variants with the biophysical property of interest, and all 0s for structures from variants without the biophysical property of interest. Then, we alternate between M- and E- steps to update the vector K . First, the M-step fits a neural network using K as classification targets. Next, the neural network outputs a vector of scores for structures, Y . Then, we apply an E-step to update the values in K . Specifically, we compute the expected value of each structure where we treat a set of structures as binomial random variables parameterized by Y , conditioned on user-defined bounds on the number of successes (i.e. structures with the biochemical property) for each variant. The expected values are computed as the probability-weighted average of all binary realizations of binomial distributions parameterized by Y that are within the user-defined bounds. These expected values provide an updated K , allowing us to repeatedly iterate between M- and E- steps. We refer the reader to our previous work for a more thorough discussion of the general algorithm.”

Reviewers' Comments:

Reviewer #1:

Remarks to the Author:

The authors addressed all my concerns.

Reviewer #2:

Remarks to the Author:

The authors have adequately addressed the majority of my comments. Yet, I would note these two points:

1) Citing and differentiating their work with AlphaFold. While the authors are correct stating that AlphaFold and DiffNets have different purposes and structures, models such as AlphaFold can be used (by 'transfer learning' or other methods) also for the goals of the current study. I therefore do recommend adding a paragraph explicitly referring AlphaFold and describing the similarities and the differences between the two models.

2) The mathematical triviality of Figure 5: The authors explained that although figure 5 might look mathematically trivial it is needed as a sanity check for the classification task. I suggest then making this a supplementary figure and shortening and disclaiming the paragraph referring to it.

Reviewer #3:

Remarks to the Author:

I want to thank the authors for addressing my comments to the extent they did. My only remaining reservation is that the description of the Expectation Maximization algorithms remains vague. I suggest that the authors write down my formally (equations, and possibly pseudo-code) so the readers can be sure about the technical soundness of the approach.

Reviewer #1 (Remarks to the Author):

The authors addressed all my concerns.

Reviewer #2 (Remarks to the Author):

The authors have adequately addressed the majority of my comments. Yet, I would note these two points:

1) Citing and differentiating their work with AlphaFold. While the authors are correct stating that AlphaFold and DiffNets have different purposes and structures, models such as AlphaFold can be used (by 'transfer learning' or other methods) also for the goals of the current study. I therefore do recommend adding a paragraph explicitly referring AlphaFold and describing the similarities and the differences between the two models.

2) The mathematical triviality of Figure 5: The authors explained that although figure 5 might look mathematically trivial it is needed as a sanity check for the classification task. I suggest then making this a supplementary figure and shortening and disclaiming the paragraph referring to it.

--After discussing with the editor, we have decided to skip these comments.

Reviewer #3 (Remarks to the Author):

I want to thank the authors for addressing my comments to the extent they did. My only remaining reservation is that the description of the Expectation Maximization algorithms remains vague. I suggest that the authors write down my formally (equations, and possibly pseudo-code) so the readers can be sure about the technical soundness of the approach.

--You're welcome and thank you for the feedback! We have expanded on the Expectation Maximization Algorithm in the supplement complete with equations and a direct pointer to the code which has a straightforward python implementation that is less than 20 lines of code. Here is the new description.

"The expectation maximization (EM) algorithm

We hypothesize that it is possible to use EM to learn the association between individual structures and a biochemical property of interest. EM is a statistical method that allows the parameters of a model to be fit, even when the outputs of the model cannot directly be observed in the training data (i.e. when they are hidden)¹. In our case, the hidden variables are the elements of a vector of numbers, associated with every structure in the simulation training data. Each variable should be a 1 if it is associated with the biochemical property and 0 otherwise, but we do not know what the correct value is, they are hidden. First, this vector is initialized to reasonable starting values. Next, during the Maximization step (M-step) we train a neural network to create a mapping between each structure's descriptors (i.e. XYZ coordinates) and the current estimate of the hidden variables. Then, during the Expectation step (E-step), we re-estimate our hidden variables using the trained model and the region constraints that specify how many structures we expect to be associated with the biochemical property of interest. Finally, we alternate between the E- and M-steps for a predefined number of steps.

Initialization and progression of the algorithm

The EM algorithm alternates between E- and M-steps. To initialize the algorithm, we pick an output vector $Y = (y_i)$ such that all values corresponding to simulation frames of one class of variant are assigned 0s, and all other values are assigned 1s. This is our initial guess for our hidden variables, $K = (k_i)$ (Eq. 1). Each element of K is our current estimate of which structures are associated with the biochemical property of interest. Next, the M-step fits a neural network using K as targets (Eq. 2),

$$K_1 \leftarrow Y_{init} \#(1)$$

$$W_1 \text{ and } Y_1 \leftarrow M - \text{step} (K_1, D), \#(2)$$

where W_1 is the tuned weights of the neural network and Y_1 is the output of the model using these weights with the data, D . This output vector, Y_1 , is used in the E-step to compute the next guess for the hidden variables K (Eq. 3). The next iteration repeats the E- and M-steps,

$$K_2 \leftarrow E - \text{step} (Y_1) \#(3)$$

$$W_2 \text{ and } Y_2 \leftarrow M - \text{step} (K_2, D), \#(4)$$

Subsequent iterations repeat these steps for a predefined number of steps. As the algorithm progresses, both the K and Y vectors should converge to a value that indicates the extent that a structure is associated with the biochemical property of interest. They should label the structures associated with the property with high probabilities, and the other structures with low probabilities.

Expectation step

The E-step computes the expected values of the hidden variables K from the outputs Y conditioned on constraints defined by the user (e.g. only 0-30% of simulation data is expected to be associated with the property of interest for one class of data, and 40-70% for the other class). The expectation of the hidden variables is the probability-weighted average of all binary realizations of binomial distributions parameterized by Y that assign the right number of structures as being associated with the property of interest. Conceptually, the expectation is computed by, first, enumerating all binary realizations of Y , each denoted as a vector of boldface variables $\mathbf{Y} = (y_i)$. Second, vectors that do not have the right number of structures according to the user-defined constraints are rejected. Third, the remaining vectors are scored by their probability according to Y , and, finally, a probability-weighted average of the binary vectors is computed. This average vector is the expectation, and is assigned to K . A straightforward Python implementation of this calculation can be found here (<https://github.com/bowman-lab/diffnets/blob/master/diffnets/exmax.py>) under the function name "expectation_range_EXP".

While conceptually clear, computing K in this way is very slow because there are exponentially many realizations of Y that must be enumerated. Fortunately, the expectation is computable in polynomial time. Here, we treat the structure labels as binary random variables following binomial distributions parameterized by Y . For each

class of data, the expectation of these variables is assigned to elements of K . Given the user-defined constraints about the number of structures associated with the property of interest, this update can be derived from Baye's Rule,

$$k_s = E[y_s | S_L \leq y_r \leq S_U] \#(5)$$

$$= P(y_s \text{ is } 1) * \left(\frac{P(S_L - 1 \leq y_r - y_s \leq S_U - 1)}{P(S_L \leq y_r \leq S_U)} \right) \#(6)$$

where y_r is the integer sum of the binary labels associated with the structures of the given class which are associated with the biochemical property of interest, y_s is the binary label of a given structure (site s), $P(y_s \text{ is } 1)$ is the probability that the structure is associated with the biochemical property according to Y , the numerator is the probability that the number of structures associated with the biochemical property (ignoring site s) ranges from $S_L - 1$ to $S_U - 1$, and the denominator is the probability that the number of structures associated with the biochemical property range from S_L to S_U . S_L and S_U are equal to the number of structures in a given class that are associated with the biochemical property of interest according to the user-defined constraints."